# LABELLING DATA WITH UNKNOWN REFERENCES

## ABSTRACT

An evaluator is trustworthy when there exists some agreed-upon way to measure its performance as a labeller. The two ways to establish trustworthiness are either by testing it, or by assuming the evaluator 'knows' somehow the way to label the corpus. However, if labelled references (e.g., a development set) are unavailable, neither of these approaches work: the former requires data, and the latter is an assumption, not evidence. To address this, we introduce an algorithm (the 'No-Data Algorithm') by which to establish trust in an evaluator without any existing references. Our algorithm works by successively posing challenges to said evaluator. We show that this is sufficient to establish trustworthiness w.h.p., in such a way that when the evaluator actually knows the way to label the corpus, the No-Data Algorithm accepts its output; and, conversely, flags untrustworthy evaluators when these are unable to prove it. We present formal proofs of correctness, empirical tests, and applications to LLMs-as-judges on low-resource languages.

## 1 INTRODUCTION

One of the most fundamental problems in AI is the question of measurement. It has been known for a while that some metrics have a certain level of unreliability; for example, by not corresponding to human judgements appropriately (Reiter, 2018; Gehrmann et al., 2023; Bavaresco et al., 2024; Hada et al., 2024). With the rise of LLMs as evaluators, typically known as LLMs-as-judges, scaling measurement to multiple subproblems has become ubiquitous (De Wynter, 2025). However, their reliability and trustworthiness as a measurement tool has led to a substantial divide in the research community, and specifically around the validity of some results (Gehrmann et al., 2023; Rabinovich and Anaby Tavor, 2025; Moghe et al., 2023).

A key component of the question of measurement, and the focus of our work, is whether the tool providing the evaluation (the *evaluator*) may be trusted. This trust is typically established with some sort of ground truth, such as an existing set of labelled references with which to perform statistical analyses (e.g., a $t$-test), or a benchmark with previously-published results. Due to uniqueness of the problem, cost of annotation, or other reasons, references may not readily exist, or be too scarce to be statistically significant. Likewise, contamination of public benchmarks has become a major concern (see, e.g., Sainz et al. 2023). It follows then that the only way one can scientifically establish trust on an evaluator whose capabilities are unknown, in a no-reference scenario, is with a formal proof.

In this paper we introduce an algorithm (the 'No-Data Algorithm') by which to establish trust in an evaluator *without* previously-labelled data. The algorithm is designed to work in-line with contemporary machine learning techniques–namely, LLMs-as-judges–where an evaluator claims it knows the map between datapoints and labels. It is up to a verifier to decide, based on the outcome of certain challenges, whether the evaluator may be trusted. The No-Data Algorithm requires a linear number of calls and **does not require labelled datasets** (e.g., development or training sets) or knowledge of the labels in the test set to establish trust.

### 1.1 CONTRIBUTIONS

Our contributions encompass theory and experiments. Theoretically, we show that the No-Data Algorithm, after $r$ calls, bounds confidence on the evaluator to a $(1/4)^r$ probability of error. This success rate reflects the evaluator's knowledge, or lack thereof. If the evaluator knows how to label the set, the accuracy and success rate, as reported by the No-Data Algorithm, will be high. Conversely,

if the evaluator lies about knowing the labels, the accuracy *could* be high–perhaps by luck–but the success rate will not.

Empirically, we test the No-Data Algorithm with traditional machine-learning (i.e., a decision tree) and contemporary (r. LLMs) evaluation methods. We also showcase an application of our algorithm to 'judging' an LLM-as-a-judge's trustworthiness when labelling a novel dataset written and annotated in West Frisian, a low-resource language. We also include ablation experiments with LLMs-as-judges, prompts, components of the algorithm, and extensions to other scenarios (e.g., $k$-ary label sets).[1] All results adjust to the theory's predictions.

Our work demonstrates that trust in an evaluator may be established via formal methods without the need for references, and enables their application in scenarios where labels may be extremely scarce, such as market research, low-resource languages (the subject of our experiments), and the medical domain. It is also, to our knowledge, the first attempt to establish trust in an LLM-as-a-judge from a mathematically rigorous perspective.

## 2 RELATED WORK

There has been considerable work on attempting to create reliable evaluators. Most of contemporary work relies on LLMs and various prompting strategies or call stacks, although the problem of (un)reliability/trustworthiness of an evaluator is older than generative models. For example, it has been known that the use of BLEU for tasks other than English-based machine translation is not accurate (Liu et al., 2016; Novikova et al., 2017); and even then, that it does not correspond well to human judgements (Reiter, 2018; Gehrmann et al., 2023).

More generally, it is known that metric choice, and hence the choice of evaluator, is system-dependent (Chen et al., 2024; Moghe et al., 2023; Flamich et al., 2025; von Däniken et al., 2024; Pombal et al., 2025). Poorly-developed rubrics also lead to unreliable human judgements (Clark et al., 2021; van der Lee et al., 2019), and thus the evaluator trustworthiness problem is also a rubric problem. There are algorithms (Northcutt et al., 2017; Zhang and Sabuncu, 2018) and paradigms (Northcutt et al., 2021) for learning with noisy teachers. Likewise, provable methods have been used with success in other areas, such as representation learning (Jovanović et al., 2023), and to improve robustness via synthetic data generation (Dimitrov et al., 2022; Fischer et al., 2022). However, they all assume that a (perhaps noisily) labelled dataset exists.

Relevant to our work, the use of LLMs as evaluators has promised extraordinary scaling capabilities, such as fast data generation and evaluation without humans (Open AI, 2023; Chiang et al., 2024; Chiang and Lee, 2023; Liu et al., 2023; Wei et al., 2024; Zheng et al., 2023). Some of these criticisms are that they do not correlate with human judgements well (Bavaresco et al., 2024; Hada et al., 2024; Chen et al., 2024; De Wynter et al., 2025; Tjuatja et al., 2024); are sensitive to their prompt (Lu et al., 2022; Hida et al., 2024; Ye and Durrett, 2022); their performance could be the product of memorisation (Lee et al., 2023; De Wynter et al., 2023; Sainz et al., 2023); and even that they do not understand the task at all (De Wynter and Yuan, 2024; Webson and Pavlick, 2022; Ye and Durrett, 2022). Their reasoning capabilities have been put into question, from the results being strongly dependent on the choice of metrics (Schaeffer et al., 2023; Chen et al., 2024), to findings that their output reasoning steps contain spurious reasoning (Turpin et al., 2023; Lanham et al., 2023; Wu et al., 2024). Most of the work on improving trust, however, relies on prompting strategies or call stacks to other LLMs (Li et al., 2024).

The main subroutine from the No-Data Algorithm, which we call the Evaluator-Verifier (EV) protocol is based off the well-known Arthur-Merlin (AM) protocol from Babai (1985). It is a type of zero-knowledge proof; see Goldreich (2009) and Arora and Barak (2009) for primers on the subject. That said, the EV protocol is different in many regards to the AM protocol, since it is designed to fit arbitrary inputs and the larger No-Data Algorithm. In particular, the EV protocol always reveals its secrets, which sets it apart from applications in theoretical computer science. Likewise, our use of the rubric and the aggregator functions are generalisations to partial functions that are not present (or needed) in previous works.

---

[1]Code and data is in `https://anonymous.4open.science/r/no_data_algorithm-C3E5`.

## 3 DEFINITIONS AND PROBLEM SETUP

### 3.1 NOTATION

Our work relates to *datapoints* $x \in X$ and *labels* $y \in Y$. Whenever the label set $Y$ is unspecified, it is the binary label set $Y = \{0, 1\}$. We also let $X \subset \{0, 1\}^n$; that is, every datapoint is a $n$-bit binary string modelling a phenomenon. The assumptions on $X$ and $Y$ are to simplify our proofs; see Section 7 for a discussion on their relevancy. For a binary string $x \in \{0, 1\}^n$, we denote its *relevant subset set* as $S_x \subset 2^x$. The construction of this specific set will depend on the setup, and will be discussed in detail in Section 4.1.

For two functions $f\colon B \to C, g\colon A \to B$ we write the composition $f \circ g\colon A \to C$ as $fg$. For two sets $A$, $B$, we denote equality **up to isomorphism** as $A \cong B$, where $A \cong B$ if $f(A) = f(B)$ but $A \neq B$ for some $f\colon U \to V$. We also consider the special case of equality **up to permutation** over subsets as $A \equiv B$, where $A \equiv B$ if $\forall a \in S_A, \exists b \in S_B \, . \, g(a) = g(b)$, for some fixed function $g\colon U \to V$. We assume all functions to be deterministic throughout, unless stated otherwise.

### 3.2 DEFINITIONS

In line with standard learning theory (Valiant, 1984), we assume that there exists an *unknown* map $f\colon X \to Y$ that takes datapoints to labels faithfully–that is, $f$ provides the true labelling. Our task is to determine whether a function $E\colon X \to Y$ (a labeller, or evaluator) is equivalent to $f$, *without* any knowledge of the map itself. This is measured with its error rate, or, conversely, its accuracy.

We also introduce four definitions which we will use in our work:

A **criterion** $c\colon X \to \{0, 1\}$ maps a datapoint $X$ to a single bit.

A **rubric** $C\colon \times_{i,\dots,n} \{0, 1\} \to \{0, 1\}^n$ maps the binary outputs from an ordered set of criteria $\mathcal{C} = \{c_1, \dots, c_n\}$ to a bitstring that is the concatenation of these outputs: $C = c_1|c_2|\dots|c_n$. We say **a rubric is total** if it explicitly decomposes nonlinear criteria with arity $> 1$ (e.g. the xor $c_i = c_a \oplus c_b$), and write it as $\tilde{\mathcal{C}}$. When the evaluation of said criteria can be done separately (e.g., by testing first $c_a$, then $c_b$, and then their xor $c_a \oplus c_b$), we say it is a **total evaluation**.

The **aggregator** $\sigma\colon \{0, 1\}^n \to Y$ is a function that maps $n$-bit binary strings (rubric outputs) to $Y$.

Finally, for two $x, x' \in X$, we say that $x$ **is similar to** $x'$ if both $x \cong x'$ and $x \equiv x'$.

Intuitively, the rubric is the set of reasons *why* a datapoint $x$ would take a given label (for example, an annotation rubric passed in to human labellers, the last layer in a model, etc.); while the aggregator decides the final label based on that rubric (e.g., via an activation function, a majority vote, etc.). Natural-language examples are in Appendix E, and discuss the need for a rubric further in Section 7.

We are now ready to introduce the core assumption of our work:

**Assumption 1.** *The map $f\colon X \to Y$ is a composition of the rubric and the aggregator:*

$$f = \sigma C. \tag{1}$$

This is not a strong assumption–after all, datapoints must have a reason to acquire a label. When there is more than one reason, there must be a way to decide how to aggregate them. However, this decomposition is crucial to our work.

## 4 THE NO-DATA ALGORITHM

The No-Data Algorithm is an algorithm with two components (players): the **evaluator** and the **verifier**. Throughout the algorithm's run, for every datapoint $x \in X$, the evaluator must convince the verifier that it is able to label $x$ correctly (i.e., that it knows $f$).

For this, the players run a multi-round sub-game which we refer to as the **Evaluator-Verifier** (EV) **protocol**. The EV protocol takes in $x$, and either succeeds or fails, w.h.p., based on challenges posed by the verifier on a $x'$ similar to $x$ generated by the evaluator. The EV protocol returns the proposed label and the status (success or failure).

The No-Data Algorithm then flips the label with probability $\phi$ in case of failure. After observing every $x \in X$, it returns the success count and final labelling. The evaluator's trustworthiness (and hence the final labelling's) is established with the success count.

The full algorithm is in Algorithm 1. We and provide a characterisation of the No-Data Algorithm (bounds, correctness, etc.) in Section 5.

---

**Algorithm 1** The No-Data Algorithm. For every $x \in X$ it calls the EV protocol to determine the trustworthiness of the label returned by the evaluator, $y = E(x)$. Based on its output, the algorithm flips $y$ w.p. $\phi$ in case of failure. Finally, it returns the success count and the labels.

---

1: **Input:** Unlabelled data $X$, evaluator $E$, verifier $V$, criterion $C$, flip probability $\phi$, rounds $r$
2: $predictions \leftarrow \{\}$
3: $successes \leftarrow \{\}$
4: **for** $x \in X$ **do**
5:      $success, y \leftarrow \text{EV}(x, E, V, C, r)$                                       // Run for $r$ rounds
6:      **if** $success$ **then**
7:          $predictions \leftarrow predictions \cup \{y\}$
8:          $successes \leftarrow successes \cup \{1\}$
9:      **else**
10:     $\tilde{y} \leftarrow \neg y$ with probability $\phi$                                     // The opposite label
11:         $predictions \leftarrow predictions \cup \{\tilde{y}\}$
12:         $successes \leftarrow successes \cup \{0\}$
13:      **end if**
14: **end for**
15: **return** $predictions, successes$

---

## 4.1 The Evaluator-Verifier (EV) Protocol

In the EV protocol both players are given a datapoint $x \in X$, rubric $C$, and nothing else. In particular, $\sigma$ is not explicitly given to either player. The evaluator *claims* to know the map $f \colon X \to Y$. Since $C$ is given, the claim is then that it has prior knowledge of $\sigma$. The verifier's goal is to be sufficiently sure that the evaluator's choice of label $x$ for a given $x \in X$ may be trusted.

### 4.1.1 Setup

The EV protocol is played for $r$ rounds. At every round, the evaluator generates a similar datapoint $x'$ to $x$, and a 'partial' label $\tilde{y}'$ that–supposedly–is equal to $y$.[2] It is partial because there is no access to $f$, and hence the best the evaluator can do is to guess $\tilde{y}'$ based on $C(x')$. Once it has generated the tuple $\langle x', \tilde{y}' \rangle$, the evaluator may not change it in this round. The verifier poses one of two challenges, selected uniformly at random. Both challenges measure equality between $x'$ and $x$, either up to isomorphism or up to permutation. If the evaluator fails, the protocol terminates and returns failure. Otherwise, it starts over. Regardless of success or failure, the protocol also returns $\tilde{y}'$. See Figure 1 for a flow diagram of the EV protocol.

The verifier challenges are central to the work, and perform both structural and valuation checks:

**Challenge 1 (Equality up to Permutation)**
**Setup:** The verifier tests the evaluator by asking it to show a similar point that leads to the same total evaluation of $C$, namely, whether $x \equiv x'$ over $\bar{\mathcal{C}}$.

**Check:** The verifier asserts that

$$\forall s \in S_x, \exists t \in S_{x'} \, . \, \forall c \in \bar{\mathcal{C}}, \, c(s) = c(t); \tag{2}$$

that is, if there is at least one substring in $x'$ matching the (total) evaluation of a criterion from $x$.

---

[2]Remark that the process by which the evaluator comes up with $x'$ does not need to be a *generation* in the LLM sense, and it is instead left up to the evaluator itself, not unlike other frameworks such as Valiant (1984).

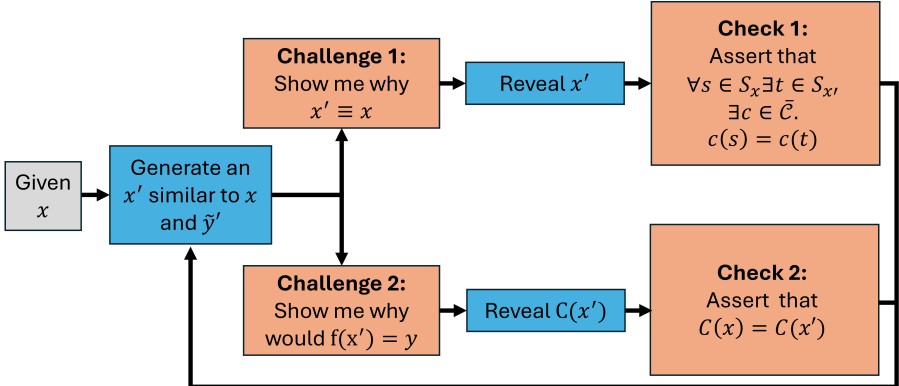

Figure 1: EV protocol flow. At every round, the evaluator (blue) generates an $x'$ similar to $x$, and a partial label $\tilde{y}'$. It then answers one of two (chosen uniformly at random) challenges by the verifier (orange). If the evaluator does not pass the challenge, the protocol returns failure. Otherwise, the game is repeated. If the rounds are over, it returns a succeeded state. In either case, it also returns $\tilde{y}'$.

**Challenge 2 (Equality up to Isomorphism)**
**Setup:** The verifier asks why $f(x') = y$. Since $y$ is not available, it checks whether $x' \cong x$.

**Check:** The verifier asserts that $C(x') = C(x)$.

**Remark 1.** *Since the challenges are public, the evaluator could cheat and generate an $x'$ to pass one of the challenges, but not both. Otherwise, the implication is that it knows $f$, and it will pass the challenges every time. This is why the challenges assert both structure ($x \equiv x'$) and final encoding ($x \cong x'$). This observation is central to our correctness proofs (Section 5).*

## 5   CHARACTERISATION

**Lemma 1** (EV Protocol Correctness Bound). *The probability that the verifier fails to detect a lie by the evaluator in the the EV protocol sub-game after $r$ rounds is $(1/4)^r$.*

*Proof.* (Sketch; a complete proof is in Appendix A.1). We separate the proof in two parts: whether the challenges are necessary and sufficient to determine whether the evaluator is lying; and the probability bound to which it happens.

On the first part, we note that Challenge 1 determines the internal structure of the string, but does not account for the valuation of $C(x')$. This challenge is necessary because two $x, x'$ could lead to the same evaluation of some $c \in \mathcal{C}$ (e.g., disjunctive clauses), and that would not mean that they are the same up to permutation.

However, Challenge 2 requires strict equality in $C(x') = C(x)$, and it is indeed order-dependent, but does not account for said internal structure. Hence both challenges provide some, but not all, information about $x, x'$ and the claimed $f(x), f(x')$. They are then both sufficient and necessary to determine if an evaluator is lying. More importantly, they are mutually exclusive in the information they provide. This is needed because a lying evaluator could know ahead of time the challenges.

For the second part we handle two cases where the evaluator could lie (or cheat): either (1) it can generate an appropriate $x'$, but not a correct $C(x')$; or (2) it provides a correct $C(x')$ but an incorrect $x'$. In either case, the probability that it gets lucky and/or passes the challenge by cheating is $1/4$; after $r$ rounds, it is $(1/4)^r$. $\qquad\square$

**Theorem 1** (No-Data Algorithm Correctness Bound). *Suppose an evaluator has accuracy $\alpha$ on a balanced dataset $X$ of size $n$ with binary labels $Y = \{0, 1\}$. Assume the evaluator always lies on any $x \in X$ iff it incorrectly labels the datapoint. Then if the EV protocol is ran for $r$ rounds, the*

*expected accuracy of the No-Data Algorithm is given by*

$$\mathrm{E}[correct] \leq 1 - (1-\alpha)\left(1 - \phi + \phi\left(\frac{1}{4}\right)^r\right), \tag{3}$$

*for a chosen value of $\phi$ and over the same dataset $X$.*

*Proof.* (Sketch; a complete proof is in Appendix A.4). The accuracy of any evaluator is directly tied to its success in the algorithm: if it knows the label, then the checks by the EV protocol will pass. Even under the constraint that the mislabelled datapoints are tied to the evaluator's lies, errors could still get past undetected. From Lemma 1, we know the error bound for every label flip, and therefore the probability of mislabelling the entire dataset. We may compute the latter via an application of the union bound, that is, $\Pr[\text{error}] \leq (1-\alpha)(1 - \phi + \phi(1/4)^r)$. $\qquad\square$

The implication of Theorem 1 is that a good evaluator with relatively high accuracy and success rate, and a well-calibrated $\phi$, will have a true accuracy higher than what the No-Data Algorithm shows. This means that reliable evaluators, as per the algorithm, may be used with the expectation that their 'true' performance will be better. Another interpretation of this result is that there is no way to recover unknowable labels when the evaluator is unable to convince the verifier.

It is worth mentioning that the assumption for Theorem 1 on the evaluator always lying when mislabelling (and hence always being wrong) is not necessarily applicable to all evaluators, since a particularly tricky evaluator could still guess the right label and yet not pass the challenges. We consider this out of scope, since the ability to guess a label without explaining how is not a viable way to establish trust, especially in scenarios without any references.

## 6   EXPERIMENTS

We present two experiments: an empirical test of the theory (Section 6.1), and an application to a realistic scenario (Section 6.2). Ablation experiments and extensions are in Appendices C.1 and D. We used two corpora: the data it knows how to label (*in-phenomenon*, or IP), and one it doesn't (*out-of-phenomenon*; OOP). We use this naming to highlight the fact that the entire measurement is altered by the change in data–and, that, semantically, we are modelling different phenomena (i.e., distinct $f\colon X \to Y$). We refer to the hypothetical setup where we know the labels of a dataset as *knowable* (r. *unknowable*). Knowable scenarios are only to illustrate how the algorithm's labelling adheres to the 'true' predictions, and in practice they would be unavailable. We run the No-Data Algorithm with $r = 3$. See Appendix F for prompts and specifics on the models and methodology.

### 6.1   EXPERIMENT 1: SYNTHETIC DATA WITHOUT NATURAL LANGUAGE

In this experiment we aimed to understand the behaviour of the No-Data Algorithm with a corpus close to that of the theoretical setup. For this, we created two disjoint synthetic datasets over binary strings. This allowed us to ablate out any potential memorisation concerns, and to control the rubrics. The rubrics are in Rubric 1. For both OOP and IP, $\sigma$ is the majority vote. The test sets for both IP and OOP included 498 entries and are balanced.

### 6.1.1   SETUP: EVALUATORS AND VERIFIERS

**Evaluators:**   We used a decision tree (DT), and the highest-performing LLM we tested (o3-mini; OpenAI 2025a).[3] Both observed a balanced training split of the IP dataset (DT was trained in it; the LLM observed it with five exemplars). We trained the DT to ensure that the evaluator learnt both IP's $\sigma$ and $C$, even though during generation it only had access to $C$, and not $\sigma$. As 'training', the LLM observed the rubric in the prompt, and was made aware of the aggregation function *only* for the labelling step. For generation it was asked to pick the best candidate from the (training) dataset instead of outputting a new point, as we observed marked performance decreases otherwise (Appendix D). During OOP, the LLM was given the rubric for IP and an OOP datapoint. We tuned $\phi$ to $\phi = 0.4, 0.1$ for the DT and the LLM, respectively. These are the worst-case scenarios, since, as per Theorem 1, it is preferable to have $\phi$ close to the knowable accuracy.

---

[3]The full results, in Appendix D, further include another closed- and two open-source models.

**Verifier:** We used a rule-matching algorithm that has access to the relevant rubrics (e.g., when testing IP, it observes the rubric for IP), implemented as described in Section 4.1.

| Rubric | Definition |
|---|---|
| **In-Phenomenon** | |
| $c_0$ | $y = 1$ if $x$ has an even number of ones; else 0 |
| $c_1$ | $y = 1$ if $x$ starts with a zero or contains $10101$, but not both; else 0 |
| $c_2$ | $y = 1$ if $x$ has strictly more than five ones; else 0 |
| **Out-of-Phenomenon** | |
| $c_0$ | $y = 1$ if $x$ contains the substring $111$; else 0 |
| $c_1$ | $y = 1$ if $x$ ends with a one; else 0 |
| $c_2$ | $y = 1$ if $x$ contains the substring $110001$; else 0 |

Rubric 1: Rubrics and criteria for our first experiment. Note that $c_1$ for the IP rubric is an exclusive-or operation: while this rubric contains three criteria, the total rubric contains five: $\{c_0, c_2, c_{1,a}, c_{1,b}, c_1\}$; where $c_{1,a}, c_{1,b}$ are the clauses for $c_1$, and $c_1$ is the exclusive-or itself.

### 6.1.2 RESULTS

The results are in Table 1. The DT and the LLM adjusted to the predictions: accuracies with the No-Data Algorithm were within $-2$ and $0\%$ of the knowable case in both IP and OOP. The success rate differences between IP (100% for DT; 81% for the LLM) and OOP (r. 5% and 28%) provided better information as to which setup was the one the evaluator truly knew. Of note, which we will cover in Section 7, is the number of flips performed by the algorithm in the DT OOP dataset: 46%.

We also performed ablation studies on the relationship of lying with evaluators and generators; the need for flipping the labels; and the effectiveness of generation strategies in various LLMs. Full details for all studies are available in Appendix D. We found that not flipping the labels led to lower accuracies in the No-Data Algorithm, especially when the evaluator lied and within IP. When the evaluator lied about knowing $\sigma$ or $f$, the success rates were lower in the IP case, but remained steady in the OOP case. The final study showed that LLM generation *under this problem* led to poor results in comparison to picking a string from a dataset. This is expected for non-natural language problems (ref. Sections 6.2 and 7) and does not impact the results.

| | **DT** (known) | **DT** (unknown) | **LLM** (known) | **LLM** (unknown) |
|---|---|---|---|---|
| **IP** | | | | |
| Successes/Flips | — / — | 100.0 / 0.0 | — / — | 81.3 / 1.8 |
| Accuracy/F$_1$ | 62.2 / 58.8 | 62.2 / 59.8 | 99.8 / 99.8 | 97.6 / 97.6 |
| **OOP** | | | | |
| Successes/Flips | — / — | 4.8 / 46.4 | — / — | 28.0 / 6.0 |
| Accuracy/F$_1$ | 54.2 / 54.2 | 52.8 / 52.1 | 60.6 / 66.2 | 59.0 / 64.0 |

Table 1: Results for the *known* (test score if the labels were known) and *unknown* (output of the No-Data Algorithm) cases. The known case is a reference that in practice does not exist. For the unknown case, if we used the base generator, the performance is exactly the same as in the known case. The flip $\phi$ is set near the error rate in the known case (0.4 for the DT; 0.1 for the LLM), since it is arguably the worst value it can have. This is needed in this experiment because the generator acts as an oracle, convincing the verifier every time, and hence the No-Data algorithm reduces to simple evaluation (the known case). However, the success rate is extremely low in DT OOP (unknown): even though there is no way for us to know whether the evaluator's score is trustworthy, the success rate indicates deception.

## 6.2 EXPERIMENT 2: LOW-RESOURCE LANGUAGE LABELLING

### 6.2.1 DATA

We created a multi-domain dataset (1,015 entries total) with randomly-drawn samples from OpenOrca (Lian et al., 2023), MMLU (Hendrycks et al., 2021), OpenCode (Ahmad et al., 2025), and WildChat (Zhao et al., 2024). It was then professionally translated and annotated into West Frisian by four native speakers. See Section 9 and Appendix F for details on the corpus creation. West Frisian is a language that, albeit spoken by half a million people, is within the class of 'exceptionally limited resources' with 'virtually no labelled data to use', as per the taxonomy from Joshi et al. (2020).

Every entry in the corpus is a pair of prompts and outputs (LLM responses from GPT-4o), and the task was to judge whether the *output* is correct based on a rubric. The rubric, written in natural language, consisted of six commonly-evaluated criteria for chatbots deployed in production, ranging from 'the response must be in West Frisian' to 'the response must be correct and without syntax errors'. For OOP we wrote a separate, almost-orthogonal rubric (e.g. 'the response must be in Dutch', as the model often responded in Dutch instead), and included a compound entry ('one of B or C must be correct, otherwise the response is zero'), bringing the total criteria to seven. Full rubrics are in Appendix E. The $\sigma$ for this corpus was label 1 if all criteria were 1, otherwise 0, and performed the evaluation with the IP labels. Remark that this is equivalent to having a lying evaluator, in line with the setup from Section 6.1.

The evaluator and verifier was GPT-4.1 (OpenAI, 2025b). We split the dataset into train (500) and test (515). Ahead of running the experiments, we calibrated the prompt strategies that worked best under the IP known case for both the final labels and the per-criterion breakdown. This is more akin to a realistic scenario, where a user would tune their verifier before comparing it in the unknown cases, and allows us to have a ground truth from which to compare the No-Data Algorithm's results.

This experiment was more difficult than the synthetic scenario, as here neither the verifier nor the evaluator knew the label *or* the criteria values, and instead both created their own label sets, and generated their own datapoints. Moreover, the rubric had some ambiguity added to it (e.g., the IP rubric requests that the model must always answer 'even if its answer is wrong'; but also that the response 'must be correct'), and only the OOP rubric was decomposable. The ambiguity was unintentional, although not uncommon to realistic scenarios. We discuss this further in Section 7 The reference labels (for the known case) are the human-provided labels under the same rubric, aggregated as a majority vote.

### 6.2.2 RESULTS

The results are in Table 2. They adjusted to the predictions, with high success rate and accuracies in the IP setting, and the converse in the OOP case. There were some nuanced differences when compared to the synthetic experiments from Section 6.1. Namely, the difference in performance was higher when $\phi$ was low (up to -3.9% and -2.2% accuracy in IP and OOP, r.). Nonetheless, success rates remained steady regardless of value of $\phi$.

It is important to note that the evaluator and verifiers in this case are the same LLM, but not the same prompt, and thus they may be considered distinct functions. Nonetheless, we performed a further ablation study comparing an evaluator model *not* proficient in West Frisian (Qwen 2.5B VL 7B; Qwen Team 2025). We found that in the IP case, where the model was given the correct rubric (i.e., it did not lie, but was wrong), the success rate was still low, hovering at around 35% in IP; and 2% for OOP (where it did lie), for all values of $\phi$. The full results are in Appendix D.4. We also performed a brief experiment when turning the rubric into a classification problem with three labels (Appendix C.1). We found that the results largely adjusted to the other results in this work, effectively spotting lying evaluators.

## 7 DISCUSSION

### 7.1 THEORY

Theorem 1 states that when the labeller has high accuracy, a low $\phi$ will be able to reproduce the label set. It is then possible to measure the successes to establish a confidence bound on the aggregate

|  | IP Acc. / F1 | OOP Acc. / F1 | IP Succ. / Flips | OOP Succ. / Flips |
|---|---|---|---|---|
| **Known** | | | | |
| Predicting $y$ | 76.3 / 80.3 | 51.5 / 67.2 | –/– | –/– |
| Avg. per $c \in \bar{C}$ | 89.5 / 94.0 | 45.3 / 46.5 | –/– | –/– |
| **Unknown** | | | | |
| GPT-4.1 | | | | |
| $\phi = 0.7$ | 72.4 / 76.6 | 49.3 / 57.8 | 87.8 / 9. 3 | 1.4 / 70.3 |
| $\phi = 0.3$ | 74.4 / 78.1 | 49.1 / 39.1 | 86.8 / 4.9 | 1.2 / 33.6 |
| $\phi = 0.1$ | 76.9 / 79.8 | 50.7 / 16.4 | 86.2 / 1.8 | 1.2 / 9.1 |

Table 2: Performance of the LLM on the natural-language version of the No-Data Algorithm. We report the tuned prompt in the known case, which in turn informed the prompts we used in the unknown case. When using the predictions for the criteria to determine the label, the accuracy of the models is 80.0% (IP), and 50% (OOP).

labels provided by the evaluator. It, however, has a different interpretation: when the accuracy of the evaluator is zero, there is no way to determine the labelling, regardless of $\phi$–it is completely unknowable. We argue that this makes sense: otherwise, it would be possible to create information out of thin air, and without any prior knowledge of $f$. Hence the primary purpose of the No-Data Algorithm is to establish trustworthiness, not to label data.

In terms of assumptions, we assumed $Y$ to be binary. This is a sufficiently strong assumption from a computational point of view, but is worth discussing further to avoid misinterpretation. For the case when $Y$ has arity $> 2$, mathematically this is easily addressable with class-based separation, at the expense of a linear-time increase in runtime. However, this is an upper bound, and we conjecture there exist versions of the No-Data Algorithm with smaller overheads. We expand upon this in Appendix C, and include a brief experimental result to highlight this conjecture.

### 7.2 EXPERIMENTS

In both experiments, the results adjusted to the theory's predictions: both the LLM and DT evaluators consistently had accuracies close to the known cases in IP/OOP, but distinct success rates in IP and OOP. This indicated that the No-Data Algorithm is robust to evaluator strategies attempting deception.

The high number of flips in the $\phi = 0.5$ DT OOP case (Section 6.1) and $\phi = 0.3$ LLM OOP (Section 6.2) support our interpretation of Theorem 1. Since the model's accuracy is near-random in the OOP dataset–as indicated by the known case–the verifier will almost always catch a lie. Hence, most labels got flipped based on $\phi$–indeed, the exact proportion marked by $\phi$–and no information was gained *from the labelling*. It is worth noting that **the success rate remained steady regardless of the value of** $\phi$, even when the flips and accuracies didn't. This means that it is an interpretable quantity, while $\phi$ is tuneable to attain 'good' labelled values.

In our synthetic data scenario (Section 6.1), we noted that LLMs struggled with generating data for that problem, and picking it from a set was more effective. However, this was not the case in the natural-language scenario (Section 6.2). We argue that this is because LLMs are primarily trained with natural language, and identification (classification) and generation are two distinct problems (Kleinberg and Mullainathan, 2024). It is thus unsurprising that synthetic problems like ours are harder for LLMs. From the perspective of our work, however, *how* an evaluator generates a datapoint makes no difference, so long as it can prove (formally) that it can be trusted to know what it is doing.

Finally, it is worth noting that the evaluation of LLMs with LLMs, as in Section 6.2, depends strongly on the LLMs themselves. This is not a circular argument *per se*. The judge (of judges) is tuned on the IP case, and asked to evaluate an unknown model in the OOP case, as done in Section 6.2 and Appendix D.4, *even when it had near-random performance in this task*. This is because they are solving different tasks, and thus the accuracies did adjust to the known cases. Moreover, distinct prompts imply distinct behaviours, and thus it is possible to treat each component separately. We provide experiments in Appendix D–and specifically, Appendix D.4–to confirm this with different

LLMs. However, from the perspective of future research directions, it raises the question on *what* are the (literature's) results measuring. We come back to this point in the next section.

## 7.3 THE NEED FOR A RUBRIC

It could be seen as limiting that the No-Data Algorithm requires a rubric, and that its performance hinges on properly designing it. We argue that this is not a (full) limitation, and, instead, a fundamental aspect of the scientific method: the verification of a hypothesis $H_0$ requires a certain set of experimental outcomes by which to decide whether to accept or reject $H_0$. These outcomes must be interpreted (i.e., evaluated as criteria), and aggregated to ultimately decide on $H_0$. Thus, the proper design of a rubric is dependent on the scientist (user), not the algorithm.

However, needing a rubric and characterising said rubric are two different things. For the latter, there are two cases that merit special attention: (a) the need for decomposability, and (b) the potential for ambiguity. For decomposability, it is worth noting that the algorithm requires these conditions for the proofs of correctness, but not for execution–the experiments from Section 6.2 use a decomposable rubric **only** in the OOP case. The No-Data Algorithm worked as intended since it is agnostic to the data labels themselves ($\sigma$ is unknown).

Ambiguity and self-contradiction could cause some loopholes, and lead to evaluators either attempting to exploit them or simply failing. We note that the rubrics from Appendix E were ambiguous, and the experiments were successful. This ties back to our argument on rejecting/accepting $H_0$: the performance of the LLM on the (human-annotated) data is evaluated with respect to the human annotations; and both depend on the design of the rubric.

## 8 CONCLUSION

We introduced an algorithm to determine trustworthiness of an evaluator in the case where labelled data does not exist. It works by performing randomly-selected checks evaluating specific parts of the datapoint to be labelled. We showed that these checks were sufficient and necessary to determine (w.h.p.) whether the evaluator knew the labels. Experimentally, the results adjusted to the theoretical predictions, with the No-Data Algorithm being able to determine lying evaluators under various scenarios, both synthetic and realistic. Of note was when the evaluator randomly guessed the labels: its accuracy was deceptively close to that of its known case, but the success rate of the No-Data Algorithm was low, thus flagging that said labelling could not be trusted. Since this quantity remained steady across all values of $\phi$, it makes it a reliable metric to determine trust without needing tuning.

One implication of the No-Data Algorithm's correctness proof is that, when the evaluator cannot be fully trusted, the data cannot be annotated. This scenario is unsolvable: there is no prior information on the data, and hence it would be effectively creating information out of thin air otherwise. This is why the algorithm's focus is on the successes: if an evaluator has low accuracy, but high success rate, it is very likely it is not lying; just wrong (e.g., the IP case in the experiments). Alternatively, high accuracy but low success rate indicates deception (r. the OOP case). Hence, **the primary purpose of this algorithm is to establish trustworthiness** and is up to the user to decide (to its own margin of trust) whether to accept or reject the labelling with respect to the rubric they designed.

We argued that establishing trust in an LLM evaluator is equivalent to establishing trust *in the prompt*, not the model, and the No-Data Algorithm was effective at this. However, since LLMs are sensitive to prompting, extensions could be needed to determine (w.h.p.) whether it is the model or the prompt which fails to convince the verifier. This is a complex mathematical problem, as the space of possible prompts is very large, but we conjecture it is not impossible. As a starting point, we note that (a) the algorithm's setup allows the verifier to 'self-calibrate', since it knows which challenges will appear; and that (b) our realistic experiment slightly departed from the theoretical work by *not* assuming that the criteria was known, and instead relied on the evaluator and verifier to generate their own judgement of both $C$ and $\tilde{y}$. It also only partially assumed decomposable rubrics, and had a small amount of ambiguity in the rubric. This is a much more difficult setup than required in the theory, yet the No-Data Algorithm performed as expected. Our work has shown that it is possible, theoretically and empirically, **to judge an evaluator's–such as an LLM-as-a-judge's– trustworthiness with formal guarantees of correctness.** Further work will expand on developing frameworks to provably ascertain trust in more evaluator classes, with a specific focus on LLMs-as-judges as described here.

## 9 ETHICS

This work's contributions involve the ability to establish trust in an LLM. We are unaware of any potential misuse of this algorithm, albeit we admit we are unable to cover all potential applications. We, however, argue that releasing the code publicly has more benefits than potential harms. For the data annotation, all annotators were contracted through an annotator services company, and compensated based on seniority, at a rate starting at 28.7 USD/hr. Annotators were encouraged to take breaks and emphasised quality over speed through the annotation instructions.

## 10 REPRODUCIBILITY STATEMENT

All code has been included as part of the supplementary materials. It will be released under the MIT licence. The data will be released on the three licences from the original works. Detailed methodology, included model versioning, is in Appendix F. Prompts are in Appendix F.1, and also in the supplementary materials. Whenever available, we set the temperature to zero to ensure further reproducibility. Given the tendency of closed-source LLM owners to update and phase out existing models, we also include experiments with open models in Appendix D. The theoretical work explicitly states the assumptions in Sections 3 and 5, and detailed proofs are in Appendix A.

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

# A    DETAILED PROOFS

In this section we provide more careful treatments of the proofs of Lemma 1 (Appendix A.1) and Theorem 1 (Appendix A.4).

## A.1    CORRECTNESS OF THE EV PROTOCOL

The key in this proof is to note that accepting a label does not mean it will be the correct label, *but* that the challenges in the EV protocol are sufficient to ascertain said correctness with some probability.

We separate our proof in two: the first part concerns itself with **completeness**, or whether the challenges will provide sufficient information to determine if the evaluator can be lying. The second part deals with **robustness** under lying, cheating, or lucky scenarios by the evaluator.

### A.1.1    PART 1 (COMPLETENESS)

We prove the following lemma:

**Lemma 2.** *Let $x, x' \in X$ be two binary strings such that $x \neq x'$. In order to test whether an evaluator knows $f(x) = f(x')$ without calling $f$, it is necessary to challenge the evaluator in both $x \equiv x'$ and $x \cong x'$. Moreover, these conditions are necessary and sufficient to determine if $x$ is similar to $x'$.*

*Proof.* (Necessary) It follows from the definition of $C$. Since $f(x) = f(x')$, testing $x \equiv x'$ is a necessary condition, but not sufficient. This challenge yields information around the internal structure of $x, x'$ up to permutation. It is necessary because it is possible to have two $x, x'$ such that $x' \notin \{\pi_i(x)\}$ (i.e., two strings that aren't permutations of one another), and yet $C(x') = C(x)$. However, this does not prove that $x$ is similar to $x'$, since there could be a subset $t \in S_{x'}. t \notin S_x$ and $c(t) = c(s)$ for some $c \in \mathcal{C}$ (for example, if $c$ contains a disjunctive clause). Hence running this challenge provides information as to whether the encoding of $C(x')$ contains all information fed into $\sigma$ for a specific $x, x'$. It accounts for variation in internal structure of the string so that $C(x) = C(x')$, but does not account for the ordering of these bitstrings.

On the other hand, $x \equiv x'$ will determine whether the encoding for the bitstrings lead to the same valuation, by checking strict equality. This is because, under our setup, the definition of similarity for $x, x'$ implies $\sigma C(x') = \sigma C(x)$. This makes it order-sensitive, but does not account for the internal structure of the string.

Since neither of them provide information about the other, it follows that both are needed to fully determine $f(x) = f(x')$.

(Sufficient) It follows directly from the earlier statement. Suppose there exists a pair $x, x'$ such that either (or both) $x \not\equiv x'$, $x \not\cong x'$ and also $f(x) = f(x')$. This would imply that (1) either the string , a contradiction.

This concludes the proof.                                                                                            □

While a natural consequence of Lemma 2 is that only checking for isomorphism is required for testing equality, we note that the goal of the challenges is to determine if the evaluator is lying, not the label itself.

### A.1.2    PART 2 (ROBUSTNESS)

We prove the following lemma:

**Lemma 3.** *The probability of error (i.e., the verifier accepts a $y' \neq y$) at the $r^{th}$ round is $(1/4)^r$.*

*Proof.* In here we have two scenarios: either the evaluator knows $f$, or it doesn't, and hence it is lying or cheating.

Knowing $f$ implies that the evaluator will generate a 'good' $x'$ every time, such that $\tilde{y}' = y$; and hence it will pass the checks from the verifier w.p.1. For the proof we focus then on the second case: that the evaluator is lying or cheating.

There are then two (disjoint) possibilities for a generated $\langle x', y' \rangle$: either that the evaluator is lying, but gets lucky; or that it is cheating and can generate the answer for one of the challenges. It is worth noting that generating an answer to pass *both* challenges is equivalent to knowing $f$. We thus focus on the two possible lies from the evaluator, which would allow it to pass either challenge.

## A.2   Lie 1: Generate a 'good' $x'$

Suppose that the evaluator generates an $x'$ such that $S_{x'} = S_x$, and thus can pass Challenge 1. It cannot pass Challenge 2, since $S_{x'} = S_x$ implies that either $C(x') \neq C(x)$ (and hence it fails it); or $C(x') = C(x)$ (and hence either it gets lucky, or it knows $f$). Since the challenges appear with probability $1/2$, the likelihood that the evaluator gets lucky in this round is $1/4$.

## A.3   Lie 2: $f(x') \neq y'$

Now suppose that the evaluator anticipates Challenge 2, and picks an $x'$ such that $C(x') = C(x)$. Similar to Lie 1, then either $S_{x'} \neq S_x$ (and thus it fails Challenge 1); or $S_x = S_{x'}$ (and hence it gets lucky or it knows $f$). The likelihood of getting lucky this round is again $1/4$.

Therefore, the probability of the verifier to *not* catch the evaluator's lie is $1/4$ in any round. At the $r^{\text{th}}$ round, $(1/4)^r$. This concludes the proof. $\qquad\square$

It follows from Lemma 2 that both challenges are sufficient to ascertain the truthfulness of a label; and from Lemma 3 that the ability of the verifier to do this is bounded. Putting both lemmas together we obtain the proof of Lemma 1.

## A.4   Correctness Bounds of the No-Data Algorithm

By a simple application of the union bound on the probability that the evaluator was wrong, but not detected by the verifier, and the probability that the evaluator was wrong, detected by the verifier, but without having its label flipped.

For this, it is easier to work with errors $\epsilon = 1 - \alpha$, and remark that, from Lemma 1 the first term (undetected, no flip) is given by

$$\Pr[\text{undetected}] = \sum_{f(x) \neq y} \left(\frac{1}{4}\right)^r, \text{ and hence} \tag{4}$$

$$\mathrm{E}[\text{undetected}] = \epsilon \left(\frac{1}{4}\right)^r. \tag{5}$$

Likewise, the second term accounts *only* for the situation where the mislabelling was detected and the label was not flipped:

$$\mathrm{E}[\text{detected, no flip}] = \epsilon \left(1 - \phi\right) \left(1 - \left(\frac{1}{4}\right)^r\right). \tag{6}$$

This yields:

$$\bigcup \Pr[\text{wrong}] \leq \Pr[\text{undetected}] + \Pr[\text{detected, no flip}] \tag{7}$$

$$\leq \left(\frac{1}{4}\right)^r + (1 - \phi) \left(1 - \left(\frac{1}{4}\right)^r\right) \tag{8}$$

$$\mathrm{E}[\text{wrong}] \leq \epsilon \left(1 - \phi + \phi \left(\frac{1}{4}\right)^r\right), \text{ by linearity of expectation.} \tag{9}$$

Substituting back $\Pr[\text{wrong}] = 1 - \Pr[\text{right}]$ and $\epsilon = 1 - \alpha$ yields the desired value. This concludes the proof.

# B  Characterisation Bounds

**Lemma 4.** *Suppose the evaluator and verifier run in polynomial time for any input $x \in X$. Then, if the No-Data Algorithm is ran with a dataset of size $|D|$ and with $r$ rounds of EV protocol, the runtime is then $O(r|D|)$.*

*Proof.* Straightforward. $\qquad\square$

**Remark 2.** *The runtime of the No-Data Algorithm is linear (up to a factor of $r$), but this algorithm is designed to calibrate an evaluator, not label a full dataset. Thus, $D$ and $r$ may be small, as in Section 6.2.*

# C  Extensions

## C.1  $k$-ary Label Sets

To convert a dataset of arity $k > 2$, it is simply a matter of following a 'one-vs-all' breakdown, rerunning the No-Data algorithm $k - 1$ times. On each run, the label set $\{l_1, \ldots, l_k\}$ is mapped as $\{l_1\}, \{l_2, \ldots, l_k\} \mapsto \{0\}, \{1\}$. In these cases, the runtime of the algorithm is increased by a factor of $k - 1$, but the theoretical guarantees hold.

It is then worthwhile considering whether it is possible to circumvent the runtime by simply running the No-Data algorithm without any modifications, except the label flip, and still obtain similar results. To test this, we ran the same setup from Section 6.2, but splitting the aggregator into three labels:

1. Label 0: All of the $c_i \in \bar{C}$ evaluate to zero.

2. Label 1: All, except one, of the $c_i \in \bar{C}$ evaluate to zero.

3. Label 2: None of the $c_i \in \bar{C}$ are zero.

The results are in Table 3. Overall, it is possible to see that the results largely adhere to the previously-observed binary label settings. Namely, lying evaluators have near-zero successes and their original performance is close to the known case, with a best-of difference of 5.8 accuracy. This difference is comparable to the binary case in Table 7 for an untuned $\phi$. Indeed can observe a slight improvement with respect to the known case when the label flip is disabled. As shown in Section 5 (and experimentally in Appendix D.3), the label flip is required to achieve full restoration of the original score, which will require further algorithmic developments for a $k$-ary setting, and are beyond the scope of this work. In sum, while empirically the No-Data Algorithm may be used for $k > 2$, further theoretical development is needed to extend this algorithm to non-'one-vs-all' strategies with the same guarantees.

We then may pose the following:

**Conjecture 1:**  Let $Y$ be a label set with arity $k > 2$, and $D$ a dataset. Then, for any number of rounds $r$, there exists an $n$ such that a version of the No-Data Algorithm runs in $O(nr|D|)$ steps, where $1 < m < n < k - 1$ and $m$ depends on $k$.

For a 'version' of the No-Data Algorithm, we mean that it maintains the same correctness bounds from Theorem 1.

## C.2  Distance-based Constraints

The constraint that $C(x) = C(x')$ is too strict for some applications. We sketch out in this section a variant of the No-Data Algorithm where, instead of requiring equality, we request closeness with respect to some arbitrary metric $\delta$ in Challenge 2. Suppose now that the failure is given whenever $\delta(C(x), C(x')) \geq \epsilon$, for some $\epsilon \in [0, 1/2]$. The bounds of correctness are roughly equivalent to these described in Lemma 1 and Theorem 1. However, the term $(1/4)^r$ now takes a dependency on the probability of generating a string that is far (based on $\delta$) from the desired constraint, and thus the bounds become weaker when said probability is larger than $1/2$.

| | IP Acc. / $F_\mu$ | OOP Acc. / $F_\mu$ | IP Succ. / Flips | OOP Succ. / Flips |
|---|---|---|---|---|
| **Known** | | | | |
| Predicting $y$ | 71.8 / 61.1 | 10.5 / 7.0 | –/– | –/– |
| **Unknown** | | | | |
| $\phi = 0.1$ | | | | |
| Flip | 66.0 / 55.7 | 12.2 / 10.8 | 39.4 / 4.3 | 0.4 / 11.3 |
| No flip | 67.0 / 57.1 | 10.1 / 6.4 | 39.4 / – | 0.4 / – |
| $\phi = 0.3$ | | | | |
| Flip | 59.8 / 52.3 | 20.0 / 17.8 | 38.1 / 18.1 | 0.8 / 34.2 |
| No flip | 61.6 / 55.1 | 10.5 / 7.0 | 38.1 / – | 0.8 / – |
| $\phi = 0.7$ | | | | |
| Flip | 39.5 / 40.0 | 30.9 / 21.1 | 35.3 / 47.4 | 0.8 / 71.6 |
| No flip | 46.8 / 43.4 | 0.10 / 6.7 | 35.3 / – | 0.8 / – |

Table 3: Ablation study on LLM generation strategies, reframing the setup from Section 6.2 into a ternary-labelled experiment. We report the results with accuracy and $F_\mu$ (macro) score. For comparison, a random guesser would obtain 33.2 accuracy and 30.1 $F_\mu$. In this experiment, the known case was still reconstructable from the algorithm's run, although with more noise. Disabling label flips improved restoration of the original scores. Full reconstruction would require the development of further algorithmic strategies to retain all guarantees from the No-Data Algorithm.

# D FULL RESULTS

## D.1 LLM-BASED RESULTS

In a full version of our study from Section 6.1, we evaluated multiple open and closed LLMs. The LLMs studied were o3-mini (as reported in the main section), DeepSeek (DeepSeek-AI et al., 2025), GPT-4o, and Qwen 2.5 VL 7B (Qwen Team, 2025), all under the same experimental setup from Section 6.1. The results can be found in Table 4. Outside of o3-mini, all models–open or not–struggled with this problem, even in the known case. Consequentially, their performance in the unknown case also adjusted to the predictions. It is worth noting that, even when their accuracy was (relatively) high in OOP, their success rate was low–as expected. This is particularly noticeable in o3-mini, GPT-4o, and DeepSeek.

## D.2 ABLATION: LYING EVALUATORS AND LYING GENERATORS

In this study we evaluated the impact of various aspects of lying in evaluation, both during generation and evaluation (of $C$). The results are in Table 5. The lies tested were when the model claims to:

1. know $\sigma$: that is, it can output an encoding $C(x)$, but does not necessarily know $\bar{C}(x)$;

2. know $f$: that is, it can retrieve a datapoint $x' \in X$ such that $y' = y$, but nothing else; and

3. approximately know $f$ up to a probability $p = 1/10$ (r. it could pass both challenges with probability $1 - p$).

The last 'lie' is more akin to how a realistic generator would behave (e.g., an LLM), sometimes understanding the challenges well, but failing them with some probability.

As before, the results indicate that lying evaluators tend to score low in the success rate, even when their accuracies are close to the known case. Remark that (a) the evaluator that lies on knowing $\sigma$ had a success rate much higher than that of lying of knowing $f$; and (b) the evaluator with approximately knowledge of $f$ had a comparatively low (44%) success rate.

This success rate depends on the cardinality of $\bar{C}$: when it is equal to $\mathcal{C}$, the model will always pass Challenge 1 (and fail Challenge 2). However, when $|\bar{C}| > |C|$, there will be criteria that the evaluator cannot possibly solve without knowing how $C(x)$ is constructed based on its internal structure, and hence it will fail them with some probability.

|  | o3-mini | DeepSeek | GPT-4o | Qwen |
|---|---|---|---|---|
| **Known Case** | | | | |
| **IP** | | | | |
| Accuracy/$F_1$ | 99.8 / 99.8 | 61.0 / 70.8 | 60.1 / 69.7 | 50.0 / 66.7 |
| Successes/Flips | –/– | –/– | –/– | –/– |
| **OOP** | | | | |
| Accuracy/$F_1$ | 59.0 / 70.0 | 54.4 / 65.6 | 55.8 / 66.4 | 50.2 / 66.8 |
| Successes/Flips | –/– | –/– | –/– | –/– |
| **Unknown Case** | | | | |
| ($\phi = 0$) | | | | |
| **IP** | | | | |
| Accuracy/$F_1$ | 54.0 / 58.9 | 46.8 / 18.0 | 42.8 / 23.6 | 54.1 / 27.3 |
| Successes/Flips | 53.8 / 46.2 | 12.9 / 87.1 | 6.6 / 93.4 | 13.3 / 87 |
| **OOP** | | | | |
| Accuracy/$F_1$ | 58.4 / 56.6 | 55.0 /39.5 | 78.7 / 78.1 | 33.5 / 0.0 |
| Successes/Flips | 26.3 / 73.7 | 16.5 / 39.5 | 34.1 / 65.9 | 16.3 / 83.7 |
| ($\phi = 0.5$) | | | | |
| **IP** | | | | |
| Accuracy/$F_1$ | 81.6 / 82.4 | 53.2 / 52.4 | 48.6 / 48.8 | 53.2 / 54.2 |
| Successes/Flips | 55.4 / 5.0 | 12.5 / 44.2 | 6.6 / 47.8 | 13.1 / 47.8 |
| **OOP** | | | | |
| Accuracy/$F_1$ | 56.4 / 59.3 | 55.4 / 58.0 | 63.3 / 67.4 | 44.4 / 54.2 |
| Successes/Flips | 28.7 / 59.3 | 19.5 / 37.8 | 33.7 / 34.1 | 16.3 / 47.8 |
| ($\phi = 0.9$) | | | | |
| **IP** | | | | |
| Accuracy/$F_1$ | 94.2 / 94.2 | 60.0 /68.4 | 56.6 / 63.8 | 51.2 / 65.4 |
| Successes/Flips | 55.4 / 5.0 | 11.9 / 9.4 | 5.8 / 10.8 | 13.1 / 8.8 |
| **OOP** | | | | |
| Accuracy/$F_1$ | 60.6 / 65.5 | 53.2 / 63.9 | 53.4 / 63.6 | 50.2 / 64.9 |
| Successes/Flips | 28.7 / 7.8 | 18.1 / 7.6 | 34.5 / 4.6 | 16.3 / 8.0 |

Table 4: Comparison of the No-Data algorithm with respect to various LLMs and values of $\phi$. The known case reflects the LLM's expected performance in the dataset, while the unknown case shows what the No-Data Algorithm outputs. In the unknown case, all LLMs broadly adjusted to their known case. The LLMs also showed on average low accuracies across the board. However, success rates were markedly different between the known and unknown cases under high accuracy, indicating possible deception. When the accuracy was low, success rates remained low regardless of the setting.

| | IP (Flips) | IP (No Flips) | OOP (Flips) | OOP (No Flips) |
|---|---|---|---|---|
| **No Lie**: DT | | | | |
| Successes / Flips | 100.0 / 0.0 | 100.0 / —- | 6.0 / 35.3 | 6.4 / —- |
| Accuracy/ $F_1$ | 60.6 / 58.8 | 59.0 / 57.4 | 48.6 / 47.3 | 45.6 / 44.8 |
| **Lie 1**: $\sigma$ unknown | | | | |
| Successes / Flips | 17.0 / 35.1 | 15.1 / —- | 2.8 / 35.1 | 2.2 / —- |
| Accuracy/ $F_1$ | 58.6 / 58.8 | 42.6 / 45.0 | 52.4 / 53.8 | 49.0 / 49.4 |
| **Lie 2**: $f$ unknown | | | | |
| Successes / Flips | 0.6 / 37.4 | 0.2 / —- | 3.0 / 35.7 | 2.8 / —- |
| Accuracy/ $F_1$ | 50.6 / 48.1 | 39.4 / 42.4 | 51.4 / 52.2 | 51.2 / 52.8 |
| **Lie 3**: generator w.p. $p = 1/10$ | | | | |
| Successes / Flips | 43.6 / 24.9 | 51.3 / —- | 4.4 / 37.9 | 4.6 / —- |
| Accuracy/ $F_1$ | 55.8 / 54.2 | 49.0 / 49.4 | 47.6 / 49.5 | 46.6 / 48.6 |

Table 5: Performance for a lying DT evaluator on the No-Data Algorithm with $\phi = 0.4$. From top to bottom: Lie 1: $\sigma$ unknown. Lie 2: $f$ unknown. Lie 3: $f$ approximately known (lying with probability $p = 1/10$) in the generation step. In all these calls, the evaluator itself also outputs the incorrect label with probability $p = 1/10$. Note the comparatively high success rates in the IP case of Lies 1 and 2. We discuss these findings in Section 7. Also note the consistently low success rate for OOP across all experiments, thus indicating the robustness of this algorithm to any evaluator strategy in scenarios where the data is completely unknown to the evaluator.

| | o3-mini (IP) | o3-mini (OOP) | GPT-4o (IP) | GPT-4o (OOP) |
|---|---|---|---|---|
| **Picking** | | | | |
| Accuracy/$F_1$ | 97.6 / 97.6 | 59.0 / 70.0 | 58.0 / 67.1 | 56.4 / 65.4 |
| Successes/Flips | 81.3 / 1.8 | 27.9 / 6.0 | 28.7 / 71.3 | 10.8 / 8.8 |
| **Generating** | | | | |
| Accuracy/$F_1$ | 94.2 / 94.2 | 60.6 / 65.5 | 56.6 / 63.8 | 53.4 / 63.6 |
| Successes/Flips | 55.4 / 5.0 | 28.7 / 7.8 | 5.8 / 10.8 | 34.5 / 4.6 |

Table 6: Ablation study on LLM generation strategies, comparing a high-performing model (o3-mini) with a lower-performing model (GPT-4o). We report the results at $\phi = 0.1$. Both LLMs were able to broadly adjust to their original accuracy, but success rates are markedly different when they are generating a datapoint versus picking it.

### D.3 ABLATION: LLM GENERATION STRATEGIES

Given LLMs' sensitivity to the prompt, it is likely that their performance in the EV protocol could be marked by *how* the datapoints were generated in the first place. For this study we tested two different generation strategies: picking a matching datapoint from the data, as in Section 6.1.2, and actually generating a datapoint from the rubric. We compared the performance of the LLM from Section 6.1 (o3-mini); and GPT-4o (OpenAI, 2024). The results are in Table 6. We found that, again, the experimental results adjusted to the predictions, with lying and low-performing models being easily spotted by the No-Data Algorithm. However, we also found that actual generation (versus picking) led to low performance for both LLMs.

### D.4 ABLATION: LINGUISTIC COMPETENCY

In this section we present an expanded version of the results from Section 6.2, including the performance of Qwen 2.5B VL 7B on the same corpus, with GPT-4.1 as a verifier. This model has no indications of being proficient in West Frisian. On the other hand, the calibration of GPT-4.1 in IP indicates that this verifier is capable of acting as a comparatively reliable component. Indeed, in the IP case, where the model was given the correct rubric (i.e., it did not lie), the success rate for the Qwen-based evaluator was low–thus indicating inability to solve the problem, as opposed to deception. Consequentially, in the OOP case (where it was set up to lie), the success rate was

|  | IP Acc. / F1 | OOP Acc. / F1 | IP Succ. / Flips | OOP Succ. / Flips |
|---|---|---|---|---|
| **Known** | | | | |
| Predicting $y$ | 76.3 / 80.3 | 51.5 / 67.2 | –/– | –/– |
| Avg. per $c \in \bar{C}$ | 89.5 / 94.0 | 45.3 / 46.5 | –/– | –/– |
| **Unknown** | | | | |
| GPT-4.1 | | | | |
| $\phi = 0.7$ | 72.4 / 76.6 | 49.3 / 57.8 | 87.8 / 9. 3 | 1.4 / 70.3 |
| $\phi = 0.3$ | 74.4 / 78.1 | 49.1 / 39.1 | 86.8 / 4.9 | 1.2 / 33.6 |
| $\phi = 0.1$ | 76.9 / 79.8 | 50.7 / 16.4 | 86.2 / 1.8 | 1.2 / 9.1 |
| Qwen 2.5B | | | | |
| $\phi = 0.7$ | 62.5 / 63.9 | 51.7 / 59.5 | 34.4 / 46.0 | 1.8 / 70.3 |
| $\phi = 0.3$ | 55.9 / 64.8 | 51.5 / 43.2 | 34.2 / 21.4 | 2.1 / 34.4 |
| $\phi = 0.1$ | 55.5 / 67.7 | 49.7 / 19.8 | 35.0 / 6.2 | 2.1 / 11.5 |

Table 7: Performance of Qwen 2.5B and GPT-4.1 on the natural-language version of the No-Data Algorithm with our ablation study. In this setup, Qwen 2.5B and GPT-4.1 act as evaluators, and GPT-4.1 as a verifier. The success rate for Qwen 2.5B was low in both the IP case (implying that the the model is not lying; just wrong) and the OOP case (where Qwen 2.5B was lying *and* wrong). Since the success rate is much lower in the latter, it follows that the No-Data Algorithm is particularly proficient at detecting confident-but-wrong evaluators.

lowest. As before, this number remained steady throughout the experimentation. The full results are in Table 7.

# E  RUBRICS

We present the full rubrics used in our experiments. Rubrics 2 and 3 have the specifications for the natural-language experiments. Both rubrics were used by the human annotators, and passed verbatim to the LLMs.

| Criterion | Definition |
|---|---|
| c1) | The response must be in West Frisian. |
| c2a) | The response must be culturally (e.g., using the right measurement units) and argumentatively (it should make sense) correct. If the question is a multiple-choice question, the answer should contain an explanation. If it requests code, it should also contain an explanation that is clear. Grammar or accuracy of the response are not measured here. |
| c2b) | The response must be correct. If it is code, it should not have syntax errors. |
| c3) | The response must be grammatically correct: coherent, good spelling, etc. Code syntax is not measured here. |
| c4) | The response must not be cut off. |
| c5) | The model must follow the instructions from the user (the prompt) exactly and completely, even if its answer is wrong. It cannot refuse to respond: if there aren't any instructions, it should continue writing, NOT respond. |

Rubric 2: Rubric for the natural-language evaluation (IP) experiment. This rubric is meant to evaluate the quality of the output of an LLM. The rubric above was given to both the human annotators and the LLM from the No-Data Algorithm.

# F  DETAILED EXPERIMENTAL METHODS

For our first experiment, we used o3-mini (version 2025-01-31) with maximum tokens set to 50,000 and left all other parameters as default. GPT-4o (version 2024-05-13) and Qwen 2.5 VL

| Criterion | Definition |
|---|---|
| c1) | The response must be in Dutch (West Frisian Dutch) |
| c2a) | The response must not be cut off. |
| c2b) | The response must make as many references as possible to Dutch culture. |
| c3) | The response must continue writing if there is no prompt. |
| c4) | (A) The response must not be cut off, or (B) The response must make as many references as possible to Dutch culture. At least one of them must be correct. If they are both zero or one, the response is zero. |
| c5) | The response must provide a summary or conclusion. It must be explicitly marked as 'summary' or 'conclusion'. |

Rubric 3: Rubric for the natural-language evaluation (OOP). This rubric was designed to be mostly orthogonal to the IP version. That meant measuring a different problem altogether, albeit with some overlaps (e.g., part of c2b overlaps with c2a in IP, c2a here is c4 in IP). Some criteria are completely different, such as c1 and c5.

7B were called with maximum tokens set to 1,024 and temperature set to zero. DeepSeek (version R1-DISTILL-QWEN-32B) also had the temperature set to zero, but the maximum tokens set to 2,048 to account for the longer reasoning trace. For the decision tree we used the implementation from sklearn[4] with all parameters, except the random seed, left as default.

We carried out the second experiment (natural language) with GPT-4.1 (version LONGCO-2025-04-14). The evaluator (labelling, criteria) and verifier were set at 256 maximum tokens, while the generator (i.e., the evaluator generating similar datapoints) was set at 5,000 tokens. All models were called with temperature set to zero.

LLMs were sometimes prone to return unparseable outputs. To mitigate this, in case of failure, we re-queried the models up to five times.

All LLMs were called via the Azure OpenAI API. All data analysis was performed on a consumer-grade laptop. Random seeds are set for the training in the code and indicated in the repository.

## F.1 PROMPTS

The prompts we used for our LLM evaluation on synthetic experiments are in Prompt 1 (evaluation) and Prompt 2 (generation). All exemplars are omitted for brevity. Whenever a parsing error occurred, we retried the call up to five times. If it failed on the fifth, we returned a random string. For the natural language experiments the prompts are in Prompt 4 (evaluator/verifier) and Prompt 5 (generator; part of the evaluator). These prompts required no exemplars. When parsing errors occurred, we retried up to five times, and otherwise returned a zero-labelled value.

[4]https://scikit-learn.org/stable/index.html

> You are labelling binary strings based on a rubric (given below).
> First return the parts of the criterion that match the string, and the values.
> Then return the label based on the aggregate function.
> # Rubric:
> <NATURAL-LANGUAGE RUBRIC GOES HERE>
> # Aggregation Function:
> Majority vote
> Return your answer in the form:
> |reasons|
> (list of reasons)
> |reasons|
> |label|
> (the label)
> |label|

Prompt 1: System prompt for the evaluator (synthetic experiment). Anchors ('|reasons|', '|label|') are used for parsing. The text contained within '|reasons|' is, as specified by the exemplars (not pictured) a chain-of-thought approach determining criterion-by-criterion matching. The rubric, explained in natural language, is inserted in the marker. Throughout the experiments we only used the IP rubric.

> You are given five datapoints and a set of criteria.
> Based on these, pick the datapoint that matches the criteria exactly.
> # Rubric:
> <NATURAL-LANGUAGE RUBRIC GOES HERE>
> # Datapoints:
> <DATAPOINTS GO HERE>
> Return your rationale, and then your chosen datapoint ONLY in the form:
> |datapoint|
> (the datapoint)
> |datapoint|

Prompt 2: System prompt for the generator (synthetic experiment). This prompt does not utilise exemplars. The rubric, explained in natural language, is inserted in the marker. Throughout the experiments we only used the IP rubric and selected datapoints from the IP training dataset, ensuring that at least one of the entries matched the criterion. Given that the model did not follow the output format exactly, some trial-and-error was needed to ensure a comparatively high parse success rate.

> You are a datapoint generator over binary strings.
> Given a rubric (given below), a datapoint, and a label, return a similar datapoint that has the same label, and fulfils the same conditions as the rubric.
> For convenience, always start with the same datapoint: 0100. It will be easier to work with.
> # Rubric:
> <NL RUBRIC GOES HERE>
> Return your rationale, and then the final datapoint in the form:
> |datapoint|
> (the datapoint)
> |datapoint|

Prompt 3: System prompt for generation used in the ablation study from Appendix D.3. Unlike Prompt 2, this prompt does take exemplars (not pictured): the first five entries in the training dataset, along with chain-of-thought-style rationale.

You are an LLM evaluator. You will be given a prompt and an response in West Frisian, meant for West Frisian readers.
Your job will be to verify if the response follows certain criteria and give a final binary score.

Check the output against the criteria below. If it fulfils the criteria, it should be a 1. Otherwise, 0. If any of the criteria score a zero, the response must be zero.

# Criteria:
<CRITERIA GO HERE>

# Output format:
<OUTPUT FORMAT GOES HERE>

Prompt 4: System prompt for the natural-language evaluator and verifier. The output format requested is always in JSON. Depending on the setup, the output format and criteria will vary. For example, if it is meant to evaluate a single criterion, say, c2b, the definition will include only the definition of said criterion and the schema will specify solely 'c2b' as a key. Remark that the verifier will always expect the criteria from the IP rubric.

You are a paraphraser evaluating a prompt and an output for an LLM.
You will be given a datapoint (prompt/output), a label, and a list of reasons why that datapoint's output has that label.
Your job will be to return a SIMILAR prompt and output, such that the OUTPUT (1) it matches the list of reasons, and (2) matches the label.
The output must match the values in the list of reasons.

Here's the rubric used for these reasons:
<RUBRIC GOES HERE>

Your response must be in JSON using the following schema:
{
    "Prompt": the new, paraphrased user prompt.
    "Output": the new, paraphrased output fulfiling the criteria.
}
Only use the keys "Prompt" and "Output"

Prompt 5: System prompt for the natural-language generator. In this scenario the prompt will always include the rubric (either IP or OOP), as labelled by the evaluator itself.

