# OpenReview forum: "Labelling Data with Unknown References"
_ICLR.cc/2026/Conference — Submitted to ICLR 2026_

### Official Review · Reviewer_kuS7 · 2025-10-25

**Soundness:** 3
**Presentation:** 3
**Contribution:** 2
**Rating:** 6
**Confidence:** 3

**Summary:**

This paper introduces the "No-Data Algorithm", a method to establish trustworthiness in an evaluator (e.g., LLM-as-judge) for labeling data without any pre-existing labeled references. The approach relies on an Evaluator-Verifier protocol, where the evaluator generates similar datapoints and partial labels, and the verifier challenges them based on structural (equality up to permutation) and valuation (equality up to isomorphism) checks using a predefined rubric and aggregator. Theoretical proofs show correctness under assumptions like decomposable rubrics and binary data/labels, and the algorithm bounds the probability of undetected lies to (1/4)^r after r rounds. Experiments validate this on synthetic binary datasets (using decision trees and LLMs) and a real-world low-resource language task (West Frisian labeling with LLMs), demonstrating that success rates distinguish knowledgeable from untrustworthy evaluators.

**Strengths:**

- The paper addresses a critical gap in evaluator trustworthiness without references, drawing from zero-knowledge proofs to create a verifiable challenge-response system. The formal proofs provide strong theoretical guarantees, bounding error probabilities in a probabilistic manner, which is a refreshing departure from empirical-only approaches in LLM evaluation literature.
- The method enables trust establishment in data-scarce domains like low-resource NLP or medical labeling, without relying on potentially contaminated benchmarks. Ablations in appendices (e.g., on generation strategies and multi-model comparisons) add depth, showing robustness across evaluators like decision trees and various LLMs (o3-mini, GPT-4o, etc.).
- Overall the paper is well-structured, with intuitive definitions (e.g., rubric and aggregator) and adequate theoretical depth.

**Weaknesses:**

- My biggest concern is that the algorithm's core assumption assumes the true labeling function f decomposes into a rubric C (criteria) and aggregator σ (e.g., majority vote); it is reasonable in theory but challenging in practice. Defining a "total" rubric that fully decomposes nonlinear criteria requires domain expertise and may not capture complex, implicit decision-making in real evaluators like LLMs, which often rely on emergent behaviors rather than explicit rules. Also, the binary string representation of data (X ⊂ {0,1}^n) and binary labels (Y = {0,1}) simplify proofs but limit direct applicability to multi-class or continuous-label tasks; while the discussion mentions extensions (e.g., via one-hot encoding), these could increase runtime by a factor of k-1 for k classes, potentially making it inefficient for high-dimensional data.
- Experiments are confined to synthetic binary strings and a niche low-resource language (West Frisian). Importantly, the EV protocol relies on evaluators generating "similar" datapoints (thus assuming high-quality generation capabilities), but ablations show LLMs struggle with this (e.g., better performance when picking from datasets vs. generating anew), raising scalability issues for complex data - where the reference data are sparse and thus No-data algorithm could actually be useful. In unknowable scenarios, tuning phi (flip probability) based on expected accuracy is heuristic and assumes some prior knowledge, contradicting the "no-data" claim.
- While results align with theory, the datasets are small (e.g., 498 entries for synthetic, 1,015 for West Frisian), and comparisons are limited. No baselines like prompt engineering or ensemble methods for trust establishment are evaluated, making it unclear if the algorithm outperforms simpler alternatives.
- The verifier needs access to the rubric, which must be provided upfront—begging the question of how to obtain a reliable rubric without data. The protocol's multi-round calls could be computationally expensive for large datasets or real-time applications, and there's no analysis of failure modes like adversarial evaluators exploiting rubric ambiguities.

**Questions:**

- How do you propose defining and validating rubrics in real-world settings where the true f is unknown? For instance, in the West Frisian experiment, how was the rubric designed and quality-controlled?

---

> ### Author Response · Authors · 2025-11-19
>
> Thank you for your comments! We have sorted them a bit to provide comprehensive answers. We apologise for the length in advance, but we have added a summary on the second comment.
>
> ## k-Class Classification
> >(...) the binary string representation of data (...) simplify proofs but limit direct applicability to multi-class or continuous-label tasks; while the discussion mentions extensions (e.g., via one-hot encoding), these could increase runtime by a factor of k-1 for k classes, potentially making it inefficient for high-dimensional data.
>
> There is a disconnect here that another reviewer pointed out (and thus we should clarify). What this algorithm evaluates is the final label/decision. I.e., we are concerned with decision problems (as in theoretical computer science; 'is this element a member of a set or not?')*, rather than (say) a real vector, such as an LLM's logits. These are theoretically feasible, but likely (as you point out) inefficient.
> Nonetheless, k-ary classification is traditionally split into one-vs-k [1], and thus in learning theory we work with binary classification problems assuming a constant (of O(k)) runtime increase. After all, there is no free lunch on this problem (more on that below)!
> Algorithmic refinements are likely possible, but left for (our) future work.
>
> *Which we did not mention in the paper, but we will!
> [1] We are unsure if links are allowed, but the Wikipedia entry on Multiclass classification has a good background on strategies to deal with the conversion of a decision problem to a k-ary decision problem.
>
> ## Experimentation
>
> >Experiments are confined to synthetic binary strings and a niche low-resource language (West Frisian). Importantly, the EV protocol relies on evaluators generating "similar" datapoints (thus assuming high-quality generation capabilities), but ablations show LLMs struggle with this (...), raising scalability issues for complex data - where the reference data are sparse and thus No-data algorithm could actually be useful.
>
> A minor clarification: the word choice ('generation') is perhaps not appropriate for an LLM era. A better verb (which we will fix) is 'proposes'. The algorithm does not care how the point is proposed/generated (could be a Chomskian grammar or a secondary decision problem), provided that the evaluator showcases understanding of the problem.
> Which is, in turn, we argue, the shortcoming of _LLMs_, as the literature has shown many times over they are persuasive, although not able to comprehend the context well.
>
> There is theoretical work backing this (which we will add); namely Kleinberg and Mullainathan, 'Language Generation in the Limit'. They indicate that generation and decision are fundamentally different problems and thus, algorithmically distinct.
>
> ## Misc
> >In unknowable scenarios, tuning phi (flip probability) based on expected accuracy is heuristic and assumes some prior knowledge, contradicting the "no-data" claim.
>
> It is worth noting that the use of phi is secondary to the success rate, and when the success rate is high, phi-tuning is unnecessary (L446)
>
> >While results align with theory, the datasets are small (e.g., 498 entries for synthetic, 1,015 for West Frisian), and comparisons are limited. No baselines like prompt engineering or ensemble methods for trust establishment are evaluated, making it unclear if the algorithm outperforms simpler alternatives.
>
> Unfortunately, it is quite difficult (and expensive) to create data that fulfils the criteria of being OOD, realistic, able to be tweaked, allowed to be releasable (consider, for example, private enterprise data); along with maintaining a relatively large pool of LLMs to test. While we agree that experimental work is small, it aligns with the theoretical framework and it is mostly used to show practicality of the work.
> That said! We are a bit unclear on how ensembling or prompt engineering could be used to establish trust on a no-label scenario. We are happy to add (small) experiments on either, but we could use some clarification on this before spending some cycles on it.

---

> > ### Author Response · Authors · 2025-11-19
> >
> > ## Rubric:
> > >My biggest concern is that the algorithm's core assumption assumes the true labeling function f decomposes (...); it is reasonable in theory but challenging in practice. Defining a "total" rubric (...) requires domain expertise and may not capture complex, implicit decision-making in real evaluators like LLMs, which often rely on emergent behaviors rather than explicit rules.
> > >(...)
> > >The verifier needs access to the rubric, which must be provided upfront—begging the question of how to obtain a reliable rubric without data.
> >
> > It is worth noting that LLM decision-making is not what it is being tested here, but rather what the scientist determines to be a 'checklist' of characteristics a datapoint should fulfil to verify a hypothesis. We will make sure to clarify this ambiguity.
> >
> > >The protocol's multi-round calls could be computationally expensive for large datasets or real-time applications, and there's no analysis of failure modes like adversarial evaluators exploiting rubric ambiguities.
> >
> > This is correct. It is hard to say if there is a way to assert trust with provably correct challenges in one shot. Indeed, all protocols within the IP complexity class rely on a polynomial (!!) number of calls. Since ours is linear, practical scalability could be left to be determined by the experimenter ('would this sample size suffice to be convinced?').
> >
> > >How do you propose defining and validating rubrics in real-world settings where the true f is unknown? For instance, in the West Frisian experiment, how was the rubric designed and quality-controlled?
> >
> > The rubric was based on an existing system--after all, a dialogue system unable to respond to these challenges could be considered to provide a poor experience.
> >
> > **In summary**, we would argue that the core weakness of the work (which we probably shouldn't admit) is the rubric _design_, not the reliance on a rubric. An experimenter must have a desiderata for verification of a hypothesis, but if said desiderata (rubric) is poorly designed, then the experiment won't work. We argue, however, that such weakness is intrinsic to the scientific process itself (half the papers on Arxiv are 'you got this result, but if we do this other thing, it doesn't work').

---

### Official Review · Reviewer_vyZn · 2025-10-30

**Soundness:** 2
**Presentation:** 3
**Contribution:** 3
**Rating:** 6
**Confidence:** 3

**Summary:**

This paper introduces the No-Data Algorithm, a theoretically motivated protocol for establishing the trustworthiness of automated evaluators, such as LLMs-as-judges, in scenarios where no labeled reference data is available. The protocol is inspired by zero-knowledge proofs and interactive challenge-response systems, allowing a verifier to probabilistically assess whether an evaluator truly knows the labeling function. Theoretical correctness and error bounds are provided, and empirical validation is conducted on synthetic tasks and a low-resource language labeling scenario.

**Strengths:**

- This paper addresses an important and challenging problem, which is how to trust automated evaluators without labeled data.
- The authors propose a novel, theoretically grounded protocol inspired by zero-knowledge proofs and interactive proofs. Additionally, this paper provides formal correctness guarantees and explicit error bounds.
- Empirical results on synthetic and real-world task (LLMs-as-judges on low-resource languages) support the theoretical claims.

**Weaknesses:**

- The proposed methods assume the existence of a shared, explicit rubric/aggregator decomposition, which may not be realistic for many annotation tasks.
- Similarity between datapoints is not well-defined for complex or unstructured domains.
- Experiments are limited to binary labels and do not cover more complex or open-ended settings. Scalability to large datasets and high-dimensional input is untested.

**Questions:**

- How can the protocol be adapted to tasks where the rubric is not easily decomposable or is inherently subjective/ambiguous?
- How is 'similarity' between datapoints defined or computed in real-world, high-dimensional, or unstructured domains?
- How robust is the method to imperfect, partial, or subjective rubrics, or to stochastic evaluators/verifiers (e.g., LLMs with temperature > 0)?
- What are the practical computational and annotation overheads for deploying the No-Data Algorithm in real-world settings?
- What does “w.h.p.” mean in the abstract?

---

> ### Author Response · Authors · 2025-11-19
>
> Thank you for your review! We will try to keep the response short.
>
> >The proposed methods assume the existence of a shared, explicit rubric/aggregator decomposition, which may not be realistic for many annotation tasks (...) How can the protocol be adapted to tasks where the rubric is not easily decomposable or is inherently subjective/ambiguous?
>
> Said rubric must be designed by the experimenter (more on that below). We would argue that the experiment in natural language does have some ambiguity/subjectivity to it. For example, L1083 ('the response must make references to Dutch culture'), could be interpreted very differently within the Netherlands than outside of it. We do note that these are very 'decisive' criteria, though, and there could be areas where this won't work (e.g., 'is this argument convincing?'). We will add this to the limitations.
>
> >Similarity between datapoints is not well-defined for complex or unstructured domain (...) How is 'similarity' between datapoints defined or computed in real-world, high-dimensional, or unstructured domains?
>
> Could you please elaborate on this? We have intentionally used binary strings to cover anything that could be represented in a computer, but that is essentially like using a rocket engine to hammer a nail. We would be happy to add in experiments/clarifications for either. We should also note that the West Frisian example is very close to a realistic scenario (but it is not the only real-world problem evaluation is applied to).
>
> >Experiments are limited to binary labels and do not cover more complex or open-ended settings. Scalability to large datasets and high-dimensional input is untested.
>
> This is correct. The algorithm is designed for decision problems (which are traditionally binary/k-ary; something we do need to clarify in the paper), and thus generation is not covered by this work. It is worth noting that, formally, generation and decision are different problems, _especially_ for LLMs (see, e.g., Kleinberg and Mullainathan; 'Language Generation in the Limit').
>
> >How robust is the method to imperfect, partial, or subjective rubrics, or to stochastic evaluators/verifiers (e.g., LLMs with temperature > 0)?
>
> Not very robust to either, but there are nuances!
> 1. The method assumes 'a' rubric, but does not specify whether the rubric should be _good_. However, as we noted in another response, this is effectively a simulation of the scientific method. An experimenter's verification of a hypothesis hinges on the experiment design ('what do you expect to observe, and how will you verify it?'). Said weakness is intrinsic to the scientific process itself--take, for example, current LLM literature and the disparate arguments in favour or against the effectiveness of (say) LLMs-as-judges.
> 2. One of our assumptions is that the original evaluation function must be deterministic. _However_, o3-mini, which is stochastic (Table 1) worked fine. We do not expect this to hold consistently, however, but this this may be considered an extension of the protocol. It is
>
> >What are the practical computational and annotation overheads for deploying the No-Data Algorithm in real-world settings?
>
> Annotation is not an overhead by design (this algorithm returns the label and the trustworthiness as part of the call). Computationally, what we have noticed is that it suffices to run this algorithm once (for $r=3$) for a relatively small dataset (about the same size of this test set) prior to deploying the evaluator itself in production. This is, however, _anecdotal_ evidence. The scalability is linear as well (as noted in the Appendix).
>
> >What does “w.h.p.” mean in the abstract?
>
> 'With high probability' -- this has a strict definition in maths: as some parameter, here $n$, goes to infinity, the probability must go to one. We will expand it in the abstract.

---

### Official Review · Reviewer_t3tA · 2025-11-01

**Soundness:** 3
**Presentation:** 3
**Contribution:** 3
**Rating:** 4
**Confidence:** 3

**Summary:**

This paper introduces the "No-Data Algorithm" for establishing trustworthiness of evaluators (including LLMs-as-judges) without requiring labeled reference data. The approach uses an Evaluator-Verifier (EV) protocol based on zero-knowledge proof concepts, where evaluators must pass challenges about datapoint similarity to establish credibility. The work provides theoretical correctness bounds and empirical validation on synthetic data and a West Frisian language evaluation task. The authors also introduce a new West Frisian dataset that is annotated by language experts and contains 1,015 items. The approach resembles semi-supervised or unsupervised based learning methods for learning.

**Strengths:**

1. The primary contribution, the "No-Data Algorithm," is a novel, formal method for establishing trust without references. Moving beyond simple correlation metrics (which require references) to a proof-based system based on a challenge-response protocol is a commendable direction. The work clearly states when labels are fundamentally unknowable and when the algorithm's purpose is establishing trust rather than producing labels.
2. Combines a synthetic binary-string setting (clean control of rubrics) with a realistic low-resource language application (West Frisian), reinforcing external validity of the theoretical predictions.The authors also create a new West Frisian dataset that is expert annotated and from a low-resourced language (1,015 items) that is a helpful contribution towards the community.
3. The findings also can help with the future research in terms of further studying the evaluator in itself since the paper makes a crucial distinction between an evaluator's accuracy and its trustworthiness. The results compellingly show that a deceptive evaluator can achieve high accuracy (e.g., in the OOP case) but will be flagged by a low success rate. This finding that the success rate is the more reliable metric for trust is a key takeaway.

**Weaknesses:**

1. The approach theoretically is an approach that could be applied in general to any classification task where an LLM judge could be utilized, but the paper has limited score in terms of experimental evaluation. In terms of benchmarking, any classification task could have been used for the paper as a potential task to compare how this approach would do against the human labels.
2. The most critical weakness is using GPT-4.1 as both evaluator and verifier in the natural language experiments (Section 6.2). This creates a circular validation where the system is essentially checking itself. While the paper acknowledges this creates questions about "what are the results measuring" (Section 7.2), this undermines the core claim of establishing independent trustworthiness. The theoretical framework assumes an independent verifier, but the implementation violates this assumption.

3. In terms of baselines for this experiment set is an algorithm that relies on supervised learning, but a strong baseline is not used for comparison in this paper but should be utilized in the next iterations. Either a classifier that is trained on the dataset, in-context learning (few shot prompting).

**Questions:**

More of suggestions for improvement.

1. Writing and the organizing of the paper needs a bit of work, there should be a set of contributions or research questions set early in the research paper, just to highlight everything in the paper and to guide the narrative. The authors contribute (or at least they’ve created) a new human annotated dataset for Frisian language but this is not highlighted until later in a section. The narrative can be easily followed if there was a clear RQ set.
2. The generalizability of the synthetic experiment is questionable. The LLM evaluator was tasked to pick a datapoint from a provided list rather than generate a new one, reportedly because generation led to poor performance. This "picking" strategy seems to be a significant concession and a deviation from the described protocol. The implications of this methodological shortcut on the validity of the results, and how this relates to the natural language experiment where generation is performed, are not sufficiently discussed.
3. It would have been helpful if there was a figure to handle the overall narrative, Fig 1 doesn’t capture the entirety of the entire research workflow proposed in the paper.
4. More on model selection for a task that is low-resourced, it would have been helpful if the impact was compared against a baseline for multi-lingual such as Cohere's Aya model (https://cohere.com/research/aya). Why was the gpt 4.1 model identified over a multi-lingual model that is trained for that language (West Firisian)?

---

> ### Author Response · Authors · 2025-11-19
>
> Thank you for your insightful comments. To keep this short, we'll address the weaknesses and some of the questions directly.
>
> > The approach theoretically is an approach that could be applied in general to any classification task where an LLM judge could be utilized, but the paper has limited score in terms of experimental evaluation. In terms of benchmarking, any classification task could have been used for the paper as a potential task to compare how this approach would do against the human labels.
>
> Indeed. Unfortunately, we also wanted to mitigate any potential memorisation concerns, which would have clouded the results (and are in-line with the reasoning of the paper: we work where labels are unknown). For this we had to create a dataset that was (almost) out-of-domain: low-resource language, with multiple tasks, and not fully translated (transcreated, instead), as well as its own annotation rubric (instead of the original, English rubric). Thus the best classification tasks are the _novel_ classification tasks... and these are expensive.
>
> > The most critical weakness is using GPT-4.1 as both evaluator and verifier in the natural language experiments (Section 6.2). This creates a circular validation where the system is essentially checking itself. While the paper acknowledges this creates questions about "what are the results measuring" (Section 7.2), this undermines the core claim of establishing independent trustworthiness. The theoretical framework assumes an independent verifier, but the implementation violates this assumption.
>
> We agree, but with a nuance: it is worth noting that the notion of independence in an LLM is more related to the prompt, rather than the model itself. Hence, the discussion in section 7.2 is related to this: trustworthiness is established with respect to the prompt, rather than the _model_. So in an LLM context, the evaluator is the LLM + prompt. This part is not discussed enough in our work (i.e., you'd have to run the algorithm multiple times to calibrate an evaluation prompt).
> Nonetheless, we also have in the appendix multiple experiments with other models (o3-mini, DeepSeek-R1, GPT-4o, and Qwen-2.5VL), as well as an extension for the natural-language problem using Qwen.
>
> > In terms of baselines for this experiment set is an algorithm that relies on supervised learning, but a strong baseline is not used for comparison in this paper but should be utilized in the next iterations. Either a classifier that is trained on the dataset, in-context learning (few shot prompting).
>
> 6.1 has a baseline trained for this specific measurement: a model (a DT) that has been trained to understand IP. If we are missing something that is implementable within the next 10 or so days we are happy to hear about it though!
>
> > Q1 Writing and the organizing of the paper needs a bit of work (...) [and] Q2: Fig 1 doesn’t capture the entirety of the entire research workflow proposed in the paper.
>
> Acknowledged; this is a common feedback which we will address.
>
> >The generalizability of the synthetic experiment is questionable. The LLM evaluator was tasked to pick a datapoint from a provided list rather than generate a new one, reportedly because generation led to poor performance. This "picking" strategy seems to be a significant concession and a deviation from the described protocol. The implications of this methodological shortcut on the validity of the results, and how this relates to the natural language experiment where generation is performed, are not sufficiently discussed.
>
> This is correct. We will further discuss it. The gist of it is that (in this specific context) the expectation of generating a $x'$ similar to $x$ is less on the production side and on the 'here, this is a point that is analogous to this one'. How the generation is performed is not within scope of the framework (although we were careful to do the picking on a separate, unlabelled dataset).
>
> >More on model selection for a task that is low-resourced, it would have been helpful if the impact was compared against a baseline for multi-lingual such as Cohere's Aya model (https://cohere.com/research/aya). Why was the gpt 4.1 model identified over a multi-lingual model that is trained for that language (West Firisian)?
>
> We picked GPT-4.1 because it was capable of returning (95%) coherent Frisian text. We have just tested Aya-101, and sadly, it was not capable of doing so. Offline tests revealed random guessing in our benchmark and less-than-grammatical outputs. When checking Aya's technical report, it appears that their data source for Frisian is only Tatoeba. Its split of Frisian is 712 sentences of around 5-7 words on average. Frisian is, sadly, such a scarce language that, outside of proprietary models, quality language models are hardly found in the wild.

---

> > ### Comment · Reviewer_t3tA · 2025-11-25
> >
> > I acknowledge the rebuttal, and thank you for the response. The concerns were addressed but my score remains unchanged.

---

### Official Review · Reviewer_f2Lc · 2025-11-09

**Soundness:** 2
**Presentation:** 2
**Contribution:** 2
**Rating:** 4
**Confidence:** 4

**Summary:**

This paper addresses a fundamental problem of trusting an evaluator (e.g., an LLM-as-a-judge) without any labelled reference data, which is increasingly common in low-resource settings, domains with expensive annotation, and scenarios where benchmark contamination is suspected. The authors propose the No-Data Algorithm, which interacts with an evaluator through a verifier using a multi-round Evaluator–Verifier (EV) protocol. The evaluator must generate similar datapoints and justifications, while the verifier issues random structural and semantic challenges. Passing both challenges over repeated rounds is provably difficult for dishonest evaluators. The algorithm incorporates probabilistic flipping of labels upon failure and yields a “success rate” metric that reflects evaluator trustworthiness.
Overall this is a well written paper and I enjoyed reading this paper.

**Strengths:**

The following are 3 strong points of this paper:
1. This paper considers a novel and timely problem. This is one of the core weaknesses of LLM-based evaluation: absence of ground truth.

2. The analytical formal guarantees presented in this paper links deception probability to the number of challenge rounds, offering provable confidence.

3. The West Frisian experiment demonstrates impact beyond toy settings and highlights where evaluator trust matters in real deployments.

**Weaknesses:**

The following are 3 weak points of this paper:
1. Semantic similarity assumption is hand-wavy in natural language and the same is acknowledged in this paper.. But the authors did not handle this properly.

2. Authors note that trust in an evaluator becomes trust in a prompt. Without robust prompting strategies, reproducibility may suffer and this undermines the practical reliability of this paper.

3. While binarisation is suggested, the mathematical treatment does not fully generalize to multi-class, hierarchical labels,continuous scoring systems.

**Questions:**

(a) Please answer the above 3 weak points.

(b) Can you please clarify what similarity means for natural language settings.. while do so, consider embedding-based constraints.

(c) Can you provide an ablation on the prompt design aspects.

(d) How easy or difficult it is to expand your approach beyond binary labels setting? O/w the impact of this work would become limited.

(e) Proving more light on understanding how dishonest evaluators are caught would improve interpretability.. and this enhances the quality of this paper.

---

> ### Author Response · Authors · 2025-11-19
>
> Thank you for your feedback! The last bit of the summary was something we particularly appreciated.
>
> Regarding your comments:
>
> >Semantic similarity assumption is hand-wavy in natural language and the same is acknowledged in this paper.. But the authors did not handle this properly (...) Can you please clarify what similarity means for natural language settings.. while do so, consider embedding-based constraints.
>
> This is accurate, although it is worth noting that the algorithm itself does not concern on the measure of similarity-as-semantics. Instead, for this algorithm, similarity is based on the evaluation of $\bar{C}$. From a theoretical perspective (L431; also Appendix C) this implies that an evaluation on an embedding-based $\bar{C}$ could be carried out by measuring distances between criteria and thresholding it, and then proceeding as usual. Assuming that the threshold is set by the experimenter, the algorithm would work as usual. We will add a clear distinction in the discussion and expand Appendix C on this.
>
> >Authors note that trust in an evaluator becomes trust in a prompt. Without robust prompting strategies, reproducibility may suffer and this undermines the practical reliability of this paper.
>
> Another way to look at this is that what this algorithm evaluates is the prompt used in the model. We do consider evaluation of the _model_ itself as an extension of this work, since it requires more robust theoretical machinery than possible. What we suspect is that calling the No-Data algorithm with various automated prompt-optimisation workflows would be sufficient, albeit expensive. Either way, further discussion is warranted in a section on limitations (which we will add).
>
> >While binarisation is suggested, the mathematical treatment does not fully generalize to multi-class, hierarchical labels,continuous scoring systems (...) How easy or difficult it is to expand your approach beyond binary labels setting? O/w the impact of this work would become limited.
>
> Rather simple, but at a cost (there is no free lunch, after all), as noted in C.2. A k-ary label setting (for k > 2) may be turned into a binary setting by performing a partition of the set (one-versus-all). This is standard in most frameworks, although the cost is a runtime increase of $k-1$.
> Other scoring systems are more complex and decidedly not (practically) achievable with this method, albeit we should note that the majority of scoring systems (including those used in LLMs-as-judges) are k-ary (although not all! This will be called out in Limitations)
>
> > Proving more light on understanding how dishonest evaluators are caught would improve interpretability.. and this enhances the quality of this paper.
>
> We will add this to the paper, thank you! Dishonest evaluators are a fundamental aspect of the work (Sec. 5, 6; Appendices A and D) but it is true that it is muddled mostly behind the proofs.

---

### Author Response · Authors · 2025-11-17

We want to take a moment to thank all the reviewers for their comments. They are all constructive and we appreciate that it is clear that all reviewers took the time to read the work.
As we work on improvements suggested by all reviewers (in particular, readability), we might comment directly to request clarifications in some points.

---

### Author Response · Authors · 2025-12-01
**Update summary**

The thing around leaks was a bit confusing--coming back from holidays and seeing these emails were a bit of a whiplash. Still, we hope the reviewers remain interested in the revised version of our work, even if they are unable to comment!

We really want to take a moment to thank the reviewers for their comments. It is quite hard to write a paper that walks the line between theory and practice _satisfactorily_. Sometimes things get handwaved, like binarisation of labels (in learning theory, everything is binary). More often than ever, theory and formal arguments are eschewed over statistical arguments (which also have their place).
The peer review process can be difficult sometimes, but we'd like to think that the comments you all provided made the paper much more robust--at least, we are very happy with the new version.

The new revision is uploaded. Broadly, the three areas that were addressed were (a) framing; (b) k-ary labels; and (c) the need for rubrics.

# Framing:
1. Clearly specified contributions and experiments done in the paper. To maintain the work within the page limit, we opted to use this subsection as opposed to a figure (but added further pointers; apologies).
2. Fixed a few vague / overlong sentences here and there (this sentence is vague on purpose--we need the space).

# Experiments
1. **k-ary labels**: We have added an experiment with k-ary (k=3) labels and further discussion on the work (mathematical and empirical).
    - We do note (here) that other inputs (high dimensional, e.g.) are covered in our work (such as by input for an LLM), as well as emphasise that this is a classification problem.
    - Other approaches (unstructured, continuous labels) are out of scope by definition, but are suggested expansions to this work. We do note that having a robust protocol to establish trustworthiness in a specific (namely, classification) class of problems is arguably better than no trust whatsoever in _any_ situation, and that binary/k-ary classification problems are frequently found in NLP.

# Discussion
1. **GPT-4.1**: We have made clear the LLM-evaluating-LLMs argument, specifically that GPT-4.1 is evaluating 'itself'--it is not. These are two distinct prompts, and there are experiments in the appendix for o3-mini, DeepSeek, 4o, and Qwen 2.5 which we will point the reader to.
    - We have also made clear the distinctions between a prompt and an LLM, which have been a source of confusion.
    - We also have clarified the section on 'what are the results measuring?'
2. **Picking/Generating**: this point needs more clarification.
    - We have added a discussion and clarification in the main body of the work, especially noting the work by Kleinberg and Mullainathan in addition to the comparison with generating in natural language (where we did not observe the issues around generating/picking, and we solely generated).
    - We have added our comments on this algorithmic perspective as well--namely, that the trust from the EV protocol stems from generating (in the mathematical sense) a similar datapoint (how this is done, it is up to the evaluator)
3. **Rubrics**: We have added discussions on the reviewer's questions (especially the section on ambiguity; it was a great question).
    - The TL;DR is: a well-designed rubric is a fundamental aspect of the scientific method. This algorithm (like any computer) _will_ do what one asks of it, and thus the measurement depends on the ability of the experimenter/scientist to define it concretely.
    - HOWEVER, some practical aspects need to be accounted for and not 'this is just maths'-handwaved. These are (in the discussion in full):
        - _Decomposability_: the rubric does not need to be decomposable (as our natural language experiment shows), but we probably should have highlighted it further (done!)
        - _Ambiguity/self-contradiction_: the rubric _should_ be ambiguity-free, but more often than ever it is not. It so happens our experiments _had_ ambiguity, which in turn reinforced our argument on the scientific method, and the fact that this algorithm is quite robust to scientist nonsense.
4. **Delineation**: The discussion had to focus on both interpretation of the theory and the experimental results, but this led to some confusion. It is now designed to be much clearer on what goes where.

---

### Meta-Review · Area_Chair_RNmG · 2026-01-02

**Summary:**

This paper proposes the No-Data Algorithm, an Evaluator-Verifier protocol inspired by cryptographic challenge-response mechanisms to establish trustworthiness of evaluators in the absence of labeled reference data. The method provides formal probabilistic guarantees and is evaluated on synthetic classification tasks and a low-resource natural language labeling setting.

The experimental evaluation violates a core assumption of the proposed framework: evaluator-verifier independence. In the natural language experiments, the same underlying model (GPT-4.1) is used as both evaluator and verifier, resulting in a form of circular or self-validation. This undermines the central claim of establishing independent trustworthiness without references. While the authors argue that trust is defined at the prompt level rather than the model level, reviewers found this insufficient to resolve the theoretical–experimental mismatch. Reviewers also raised concerns about strong assumptions on rubric decomposability and datapoint similarity in realistic, high-dimensional domains, as well as limited empirical validation (small datasets, narrow tasks, and deviations from the proposed protocol such as picking rather than generating similar datapoints). These issues reduce confidence in the practical generality and robustness of the approach.

Due to concerns regarding a key theoretical assumption and remaining methodological issues, the paper is recommended for rejection.

**Reviewer Concerns:**

Reviewer t3tA: The rebuttal acknowledged concerns about framing, clarified the distinction between accuracy and trustworthiness, discussed prompt-level interpretation of evaluators, and expanded discussion on experimental choices (e.g., picking vs. generating similar datapoints). **The key concern regarding evaluator-verifier independence remains unresolved.**

Reviewer f2Lc: The rebuttal clarified the handling of k-way labels, expanded discussion on similarity in natural language settings, and provided additional explanation on how dishonest evaluators are detected. **Concerns regarding the practical meaning of semantic similarity and the reproducibility implications of prompt dependence remain only partially addressed, with limited empirical validation.**

Reviewer vyZn: The rebuttal responded to questions on rubric ambiguity, stochastic evaluators, and computational overhead. **The reliance on explicit, experimenter-defined rubrics and the lack of validation in complex or unstructured real-world settings remain open concerns.**

Reviewer kuS7: The rebuttal clarified the focus on decision problems, explained extensions to multi-class settings, and discussed the role of “generation” versus “proposal” of similar datapoints. **Concerns about scalability, practical rubric construction without data, and limited experimental breadth remain insufficiently resolved.**

**Reviewer Scores:**

All the score may not be changed.

---

### Decision · Program_Chairs · 2026-01-26

Reject